# Polysaccharide from *Atractylodes macrocephala* Koidz Binding with Zinc Oxide Nanoparticles as a Novel Mucosal Immune Adjuvant for H9N2 Inactivated Vaccine

**DOI:** 10.3390/ijms25042132

**Published:** 2024-02-09

**Authors:** Xiaopan Liu, Xinyi Lin, Hailong Hong, Jing Wang, Ya Tao, Yuying Huai, Huan Pang, Mingjiang Liu, Jingui Li, Ruonan Bo

**Affiliations:** 1College of Veterinary Medicine, Jiangsu Co-Innovation Center for Prevention and Control of Important Animal Infectious Diseases and Zoonoses, Yangzhou University, Yangzhou 225009, China; lxp19940@163.com (X.L.); 19822615727@163.com (X.L.); 15397890265@163.com (H.H.); jwang421@163.com (J.W.); taoya0902@163.com (Y.T.); 2005huaiyuying@163.com (Y.H.); mjliu1@163.com (M.L.); 2Joint International Research Laboratory of Agriculture and Agri-Product Safety, The Ministry of Education of China, Yangzhou University, Yangzhou 225009, China; 3School of Chemistry and Chemical Engineering, Yangzhou University, Yangzhou 225009, China; panghuan@yzu.edu.cn

**Keywords:** AMP-ZnONPs, oral, mucosal immunity adjuvant, intestinal sIgA response, gut microbiota

## Abstract

H9N2 avian influenza poses a significant public health risk, necessitating effective vaccines for mass immunization. Oral inactivated vaccines offer advantages like the ease of administration, but their efficacy often requires enhancement through mucosal adjuvants. In a previous study, we established a novel complex of polysaccharide from *Atractylodes macrocephala* Koidz binding with zinc oxide nanoparticles (AMP-ZnONPs) and preliminarily demonstrated its immune-enhancing function. This work aimed to evaluate the efficacy of AMP-ZnONPs as adjuvants in an oral H9N2-inactivated vaccine and the vaccine’s impact on intestinal mucosal immunity. In this study, mice were orally vaccinated on days 0 and 14 after adapting to the environment. AMP-ZnONPs significantly improved HI titers, the levels of specific IgG, IgG1 and IgG2a in serum and sIgA in intestinal lavage fluid; increased the number of B-1 and B-2 cells and dendritic cell populations; and enhanced the mRNA expression of intestinal homing factors and immune-related cytokines. Interestingly, AMP-ZnONPs were more likely to affect B-1 cells than B-2 cells. AMP-ZnONPs showed mucosal immune enhancement that was comparable to positive control (cholera toxin, CT), but not to the side effect of weight loss caused by CT. Compared to the whole-inactivated H9N2 virus (WIV) group, the WIV + AMP-ZnONP and WIV + CT groups exhibited opposite shifts in gut microbial abundance. AMP-ZnONPs serve as an effective and safe mucosal adjuvant for oral WIV, improving cellular, humoral and mucosal immunity and microbiota in the gastrointestinal tract, avoiding the related undesired effects of CT.

## 1. Introduction

H9N2 avian influenza virus is one of the major subtypes circulating in poultry populations worldwide [1]. Some studies suggest that this virus may have the potential to cause a more severe illness in humans [2,3]. In addition, infection with H9N2 has caused significant economic losses for the poultry industry [4]. Therefore, the prevention of the H9N2 influenza virus is an important strategy to control potential future epidemics or pandemics.

Injection is a common method of vaccination [5]; however, many pathogens invade the body via the mucosal route [6]. Therefore, the development of vaccines and adjuvants that target mucosal immune responses is particularly critical. Compared to injectable vaccines, oral vaccines have the advantages of easy administration, improved mucosal immunity and no needle contamination [7]. Cholera toxin (CT) has been widely used as a mucosal adjuvant due to its potent immunostimulatory properties. However, there are some disadvantages to using CT as an adjuvant, such as the fact it causes severe diarrhea and dehydration, even death in some cases, it is unstable under certain conditions and the production of cholera toxin is expensive [8].

*Atractylodes macrocephala* Koidz (AM) is an important traditional Chinese medicine. The polysaccharide from AM (AMP), as one of the main active components, has anti-inflammatory [9], antioxidant [10] and immunomodulatory activities [11]. Concurrently, zinc oxide nanoparticles (ZnONPs) exhibit immunomodulatory properties, capable of activating dendritic cells, leading to the production of cytokines such as IL-6 and TNF-α [12]. In the acidic environment of the small intestine, ZnONPs predominantly exist in the form of ZnOH^2+^ [13]. This characteristic is pivotal as it facilitates the attraction of ZnONPs to negatively charged cell membranes, enabling the transportation of particles into intestinal cells. Previous research has explored the binding of AMP to ZnONPs (AMP-ZnONPs) and its potential application as a vaccine adjuvant. This complex was demonstrated to activate macrophages through the TLR4/MyD88/NF-κB signaling pathway [14], thereby enhancing immunity.

We hypothesize that the distinctive electrostatic properties of ZnONPs may enhance the uptake of AMP-ZnONPs by intestinal immune cells compared to AMP alone. In order to validate our hypothesis, mice received the different formulations of vaccine orally on the 0th and 14th days, various samples were collected to measure antibody levels, immune cell activity and mucosal immune effects. Additionally, the changes in the intestinal flora induced by the vaccine were also evaluated. The findings of this study may provide valuable insights into the development of inactivated H9N2 vaccines for the oral route in the future.

## 2. Results

### 2.1. Specific HI, IgG, IgG1 and IgG2a Antibody Levels

Compared to the control group, the weight of mice in the WIV + CT group decreased significantly 28 and 35 days after primary immunization (Figure 1B); this phenomenon indicated an adverse reaction to CT in mice. Excitingly, the mice in the WIV + AMP-ZnONP group showed weight gain compared to the control group, indicating that AMP-ZnONPs are relatively safe as mucosal adjuvants.

The HI antibody titers (Figure 1C), IgG, IgG1 and IgG2a levels (Figure 1D) in the WIV + AMP-ZnONP group were increased compared to those in the control, WIV, WIV + AMP and WIV + ZnONP groups (*p* < 0.05), indicating that the AMP-ZnONP adjuvant had a stronger humoral immune response than the WIV, AMP or ZnONPs alone. The HI antibody titer, IgG, IgG1 and IgG2a levels of mice in the WIV + AMP-ZnONP group reached their highest level 28 days after primary immunization.

### 2.2. Differentiation between CD4^+^ and CD8^+^ T Cells

To further investigate the adjuvant activities of AMP, ZnONPs, AMP-ZnONPs and CT, the activation of CD4^+^ and CD8^+^ T lymphocytes from the spleen was measured (Figure 2A). The AMP-ZnONP adjuvant-based vaccine promoted the percentages of CD4^+^/CD3^+^ T lymphocytes and CD8^+^/CD3^+^ T lymphocytes (Figure 2B), which were higher than those in other groups 28 days after primary immunization. The results suggested that the AMP-ZnONPs were more effective in activating CD4^+^ and CD8^+^ T lymphocytes than the other adjuvants (AMP and ZnONPs).

### 2.3. The mRNA Expression Levels of Th1 and Th2 Cytokine

The mRNA expression of cytokines TNF-α and IL-6 was significantly higher in the jejunum of mice immunized with WIV + AMP-ZnONPs compared to those that received only WIV or PBS (Figure 2C). On the other hand, compared to WIV + AMP and WIV + ZnONPs, the IL-6 mRNA level exhibited a significant difference in the WIV + AMP-ZnONP group.

### 2.4. The sIgA Level at Mucosal Sites

The sIgA is a type of antibody produced in mucosal tissues. The WIV + AMP-ZnONP group showed the highest increase in sIgA secretion compared to the WIV, WIV + AMP and WIV + ZnONP groups (Figure 3A). Furthermore, compared with the control group, secretions of the α-chain (Figure 3B) and J-chain (Figure 3C) in the WIV, WIV + AMP, WIV + ZnONP and WIV + AMP-ZnONP groups were significantly increased to promote the mucosal immune function. The results indicated that the combination of AMP and ZnONPs had a synergistic effect in promoting small intestinal mucosal immunity by increasing sIgA secretion in the small intestinal mucosa.

### 2.5. The Change in IgA-Secreting Cell Areas in the Intestinal Mucosa

IgA^+^ cells are predominantly located at the core of the intestinal villi and lamina propria. These cells exhibit brown dots and their appearance is circular or elliptical with sparse cytoplasm [15,16,17]. Compared to the control group, there was no significant change in the IgA-secreting cell areas in the WIV group (Figure 3D,E). In addition, compared with the WIV, WIV + AMP and WIV + ZnONP groups, the WIV + AMP-ZnONP group significantly increased IgA-secreting cell areas in the duodenum (Figure 3F) and the jejunum mucosa (Figure 3G).

### 2.6. Effects of Oral Immunization on Dendritic Cells (DCs) or B Cells in the Spleen, Intestinal PP and MLN

The results of IHC and ELISA experiments (Figure 3) showed that oral administration of AMP-ZnONPs as a mucosal adjuvant significantly increased the expression level of sIgA, indicating its potential importance as a new strategy to enhance mucosal immunity. However, the underlying mechanisms by which AMP-ZnONPs regulate sIgA production remain to be fully elucidated. Therefore, we first examined the impact of AMP-ZnONPs on DCs and B-cell subpopulations in mice 28 days after primary immunization.

Compared to the control group, the DCs in the PP and spleen were significantly increased in the vaccine group supported with AMP-ZnONP adjuvant (Figure 4A and Figure 5A). It was noteworthy that the number of DCs in the MLN was less than 2% (Figure 6A), and there was no significant difference compared to the control group, WIV, WIV + AMP and WIV + ZnONP groups (Figure 6B). Thus, it could be inferred that DCs in the PP played a crucial role in mice 28 days after primary immunization in the WIV + AMP-ZnONP group. Although the number of DCs in the MLN was less, they still contributed to the immune response post vaccination, indicating that the effect of AMP-ZnONPs on DCs varies depending on different tissue types.

Double staining with CD5 and B220 can distinguish between B-1 and B-2 cells. B-1 cells are defined as CD5^+^ B220^+^ and B-2 cells as CD5^−^ B220^+^ [18]. The AMP-ZnONP adjuvant group significantly increased the populations of B-1 cells compared to all the other groups (Figure 4B). Furthermore, the AMP-ZnONP adjuvant group exhibited a remarkable increase in the number of B-2 cells compared to the other groups, except for the WIV + CT group (Figure 5B). These results suggested that CT primarily affected B-2 cells, while the AMP-ZnONPs notably influenced B-1 cells.

### 2.7. Relative mRNA Expression Levels of Intestinal Mucosal Immune-Related Gene

After the oral administration of the WIV + AMP-ZnONP vaccine, significant increases in the mRNA expression levels of key mucosal immune molecules (pIgR, APRIL, BAFF, CCR9 and CCL25) were observed in jejunal tissues compared to the control group and WIV group (*p* < 0.05). However, there was no significant difference in the BAFF expression between the WIV + AMP-ZnONP and WIV + CT groups (Figure 6C). These results suggested that CT might primarily affect the production of sIgA through its regulation of BAFF expression, whereas AMP-ZnONPs could intervene in this process by modulating the expression of multiple critical molecules, such as pIgR, BAFF, APRIL, CCR9 and CCL25. Overall, the results provided evidence that AMP-ZnONPs had a broader and more diverse effect on the activation of intestinal B cells compared to CT, which might contribute to their superior adjuvant activity.

### 2.8. Effect of AMP-ZnONPs on Intestinal Flora in Mice

The total number of raw reads obtained from this analysis was 1,509,732 (Table 1). A flat rank abundance curve, dilution curves and a species accumulation boxplot (Figure 7A) suggest reliable sample extraction and that sequencing has covered most of the bacteria in the environment.

#### 2.8.1. Alpha-Diversity and Beta-Diversity

To further analyze the effects of WIV + AMP-ZnONPs on mouse gut microbiota richness and diversity, alpha diversity analysis was performed to obtain the Chao1 index, Shannon index and observed species index (Figure 7B). The Shannon index and observed species index of the WIV + AMP-ZnONP group differed significantly from those of the control group. The WIV + AMP-ZnONP group had higher values for the Chao 1 index compared to the WIV group, and these differences were statistically significant.

PC1 accounted for 75.55% of the variation in these six groups, while a 5.53% effect was observed in PC2 according to the vertical axis (Figure 7C). The closer the distance between the groups, the more similar the gut microbiota structure [19]. The results revealed that the cluster representing the control group was distinctly separated from the WIV + AMP ZnONP group. In contrast, the control group exhibited closer clustering with the WIV and WIV + CT groups, suggesting that WIV + AMP ZnONPs exerted a significant impact on the composition of the mouse gut microbiota, whereas the microbiota in the WIV and WIV + CT groups remained relatively unchanged.

#### 2.8.2. Microbial Community Composition

The community composition of samples is similar as they cluster together in the UPGMA clustering tree [20]. In Figure 8A, a notable distance exists between the control, WIV + CT and WIV + AMP ZnONP groups. This observation aligns well with the findings from the PCoA, further corroborating the significant differences in microbial composition among these groups.

At the phylum level (Figure 8B), changes in the gut microbiota structure were observed after WIV + AMP-ZnONP intervention, with *Firmicutes* remaining dominant and increasing in number, while the number of *Firmicutes* in the WIV + CT group showed a decreasing trend. Meanwhile, the abundance of *Bacteroidetes* and *Verrucomicrobiota* in the WIV + AMP-ZnONP group showed a decreasing trend compared with the control group. Interestingly, the WIV + CT group displayed an increasing trend in *Bacteroidetes*. At the family level (Figure 8C), compared to the control group, both the WIV and WIV + AMP-ZnONP treatment showed an increased abundance of *Lactobacillus* (a family within the *Lactobacillus* group), *Oscillospiraceae* and *Lachnospiraceae* in the intestinal microbiota. Also, the WIV + AMP-ZnONP group exhibited significant changes compared to the WIV group.

#### 2.8.3. Differential Intestinal Flora between the Subgroups of WIV and WIV + AMP-ZnONPs

Furthermore, the difference in gut microbiota composition between the WIV and WIV + AMP-ZnONP groups was assessed using LEfSe (LDA Effect Size) (Figure 8D,E). In the fecal samples from the WIV + AMP-ZnONP group, the genera *Oscillospiraceae*, *Clostridia*, *Marinifilaceae*, *Eubacterium*, *RF39* and *Lachnospiraceae* were found to be enriched. On the other hand, the fecal samples from the WIV group showed the enrichment of the genera *Erysipelotrichales*, *Staphylococcus*, *Sporosarcina_pasteurii*, *Rikenellaceae*, *Staphylococcates*, *Ligilactobacillus*, *Lactobacillus_murinus* and *Bacteroides*.

#### 2.8.4. Functional Prediction of Intestinal Microbiota

The results showed that strains with significantly increased abundance in the WIV + AMP-ZnONP group mainly contribute to organismal systems and cellular processes (Figure 8F). Conversely, strains with significantly increased abundance in the WIV + CT group primarily participate in human diseases, metabolism and genetic information processing. However, no significant difference in microbiota abundance was observed among the WIV + ZnONPs, WIV + AMP, WIV and control groups.

## 3. Discussion

Vaccination is the main strategy to control H9N2 outbreaks in the poultry industry in China. However, mucosal immunization by oral delivery with an inactivated virus alone is not sufficiently effective. Adjuvants can stimulate the immune system to produce a stronger response against the H9N2 virus. Therefore, the use of adjuvants in oral H9N2 virus vaccines can improve their efficacy. In this experiment, to study the mucosal adjuvant activity of AMP-ZnONPs, classical mucosal adjuvant CT was used for positive control and oral immunization was performed on mice.

The HI assay is used to assess the immune response of vaccinated animals, and the IgG, IgG1 and IgG2a levels in the blood can be used to reflect the strength of the immune response against a specific antigen or pathogen [21]. In this study, compared with other groups, higher specific antibody levels (Figure 1C,D) in the WIV + AMP-ZnONP group indicated a more robust immune response [22].

The spleen is an important organ of the immune system that plays a crucial role in the development and function of T cells [23]. CD3 proteins are essential for the activation of T cells. CD4^+^ T cells and CD8^+^ T cells play important roles in the immune system by coordinating and executing the immune response against pathogens. Our study found that AMP-ZnONPs increased the proportion of CD4^+^ T cells and CD8^+^ T cells in the spleen, in line with the results reported by Gu et al. [24]. The result suggested that AMP-ZnONPs enhanced the cell-mediated immune response. The activation of CD4^+^ T cells was differentiated into various subpopulations, including Th1 and Th2 cells, and played an important role in coordinating immune responses to pathogens [25]. In this study, the mRNA expression levels of IL-6 and TNF-α significantly increased in the WIV + AMP-ZnONP group. Th1 cytokines like TNF-α are crucial for cell-mediated immunity, aiding in the elimination of intracellular pathogens. Th2 cytokines such as IL-6 enhance humoral immunity [26]. The RT-qPCR results of IL-6 and TNF-α (Figure 2C) indicated that WIV + AMP-ZnONPs could stimulate Th1/Th2 responses in immunized mice.

SIgA plays a critical role in protecting these mucosal surfaces from invading pathogens [27]. SIgA is formed by the combination of the J-chain and α-chain [28], and these two components are essential for the function and transport of sIgA in the immune system’s defense against pathogens at the mucosal surface. In this study, the results of qRT-PCR (Figure 3A–C) showed that the oral administration of AMP-ZnONPs as an adjuvant significantly increased the expression level of sIgA, which was consistent with the findings of the CT adjuvant group, as confirmed by the ELISA assay and IHC results (Figure 3D–G). So, what factors influence the expression of sIgA? In PP and MLN, DCs can present unprocessed antigens to B cells [29] and induce them to express the intestinal homing receptor CCR9 for antibody-secreting cells (ASC) [30]; when CCR9 interacts with the ligand CCL25, ASC can migrate towards the MLN and PP. Activated B cells can differentiate into IgA antibody-secreting cells (IgA-ASC) and can secrete IgA via T-cell-dependent (TD) and T-cell-independent (TI) pathways, with TI antigens activating only B-1 cells and TD antigens activating B-2 cells [31,32]. Our results showed that the AMP-ZnONP adjuvant group significantly increased the populations of DCs, B-1 and B-2 cells to varying degrees. CT primarily affected B-2 cells, while the AMP-ZnONP adjuvant notably influenced B-1 cells (Figure 4B and Figure 5B). A recent study demonstrated that a type 1 IFN-inducing adjuvant promotes IgA response by increasing the number of DCs to protect mice from influenza virus infection [33]. Interestingly, the study by More et al. found that the co-culture of B-T-DC cells in vitro could promote the formation of sIgA [34]. Therefore, we inferred that the AMP-ZnONP adjuvant could impact the DC populations in the PP and the spleen, as well as the TD and TI pathway in MLN and PP, but it seemed to focus more on the TI pathway, consequently influencing the secretion of sIgA. In addition, the DCs can secrete BAFF or APRIL to regulate Ig class switching in the B-cell response [35]. Following this, pIgR on the surface of intestinal epithelial cells can bind to IgA and facilitate its transport across the mucosal epithelial barrier and into the lumen of the mucosa [36], which is secreted into the intestinal lumen. In this experiment, the relative expression levels of intestinal mucosal immune-related gene mRNA in the WIV + AMP-ZnONP group were significantly increased compared to the control and WIV groups, suggesting the activation and recruitment of intestinal B cells. In line with this, Grasset et al. provided evidence that the production of sIgA responses to symbiotic bacteria in the gut was mediated by the BAFF/APIL pathway, which requires the involvement of the TACI receptor on B cells [37].

The intestinal microbiota is essential for maintaining intestinal homeostasis, the response of intestinal mucosal IgA to vaccines and immune system development [38]. Therefore, we collected fecal samples from mice 28 days after primary immunization and analyzed the composition of the intestinal microbiota. The alpha diversity analysis showed that the diversity of species in the WIV + AMP-ZnONP group was higher than those in the control and WIV groups, indicating a healthier gut microbiota. The UPGMA clustering tree and the PCoA results further corroborated the significant differences in microbial composition among these groups. To reveal the species composition of the gut microbiota, the abundance of bacteria at different levels was evaluated. The number of *Firmicutes* was increased in the WIV + AMP-ZnONP group at the phylum level, possibly due to its ability to metabolize indigestible polysaccharides into short-chain fatty acids [39]. Compared with the control group, the WIV + CT and WIV + AMP-ZnONP groups exhibited contrasting trends in the abundance of *Firmicutes* and *Bacteroidetes*. These alterations in microbiota are likely associated with CT-induced weight loss. At the family level, AMP-ZnONPs caused an increased abundance of *Lactobacillus*, *Oscillospiraceae* and *Lachnospiraceae* in the intestinal microbiota. Research has shown that certain strains of *Lactobacillaceae* may promote the secretion of sIgA, thereby enhancing immune defense in the intestinal mucosa [40]. The bacteria of the genus *Oscillospiraceae* and *Lachnospiraceae* are important members of the intestinal microbial community. They are involved in the degradation of complex sugars, the production of short-chain fatty acids and the regulation of gut inflammation [41]. To investigate the functional changes of the gut microbiota, Tax4Fun was utilized for functional prediction. It is worth noting that cellular immunity in the WIV + AMP-ZnONP group involved various cellular processes such as antigen presentation, which played a role in assisting B cells to produce sIgA [37,42], while the gene function of intestinal flora associated with human diseases was increased significantly in the CT group. These findings suggested that AMP-ZnONPs might improve gut microbiota structure and enhance vaccine immunity through these microorganisms. This study has delved into how the AMP-ZnONP adjuvant regulates sIgA secretion by affecting immune cells (DCs, B-1 and B-2 cells) and mucosal immune factors (BAFF, APRIL, CCR9 and CCL25). In addition, the AMP-ZnONP adjuvant modulates intestinal flora without the side effects associated with CT. However, to gain a deeper understanding mechanism of action of AMP-ZnONP adjuvants in immunomodulation, further studies on their complex interactions with the gut microbiota and sIgA secretion are necessary.

## 4. Materials and Methods

### 4.1. Reagents

H_2_O_2_ and CT were obtained from Sigma-Aldrich (St. Louis, MI, USA). Anti-mouse CD3-APC, CD4-FITC, CD8-PE, CD11c-eFluor 450, MHCII-APC, CD5-FITC and B220-PerCP-eFluor™ 710 antibodies were bought from eBioscience (San Diego, CA, USA). Horseradish Peroxidase (HRP)-conjugated goat anti-mouse IgA, HRP-conjugated goat anti-mouse IgG, HRP-conjugated goat anti-mouse IgG1 and HRP-conjugated goat anti-mouse IgG2a were obtained from Southern Biotech (Birmingham, AL, USA). RNA-easy Isolation was purchased from Vazyme Biotech Co., Ltd. (Nanjing, Jiangsu Province, China). SYBR qPCR Master Mix for quantitative real-time PCR (qRT-PCR) was bought from Yeasen Biotechnology Co., Ltd. (Shanghai, China). TMB was purchased from Beyotime Biotechnology (Shanghai, China). Red blood cell lysis buffer was purchased from Solarbio Science & Technology Co., Ltd. (Beijing, China). Silane coupling agent (KH550) and ZnONPs (purity 99.9%) were the products of Sinopharm Chemical Reagent Ltd. (Shanghai, China). The purified AMP was prepared in our laboratory (Yangzhou, Jiangsu Province, China). H_2_SO_4_, alcohol and HCl were obtained from Hushi Laboratory Equipment Co., Ltd. (Shanghai, China). Elisa plate was bought from cellpro biotechnology Co., Ltd. (Suzhou, Jiangsu Province, China).

### 4.2. Antigens

H9N2 virus (A/chicken/XZ/508/2019) was donated from the Key Laboratory of Jiangsu Preventive Veterinary Medicine, Key Laboratory for Avian Preventive Medicine, Ministry of Education, College of Veterinary Medicine, Yangzhou University. EID_50_/0.1 mL = 10^−8.83^. H9N2 antigen was collected from the allantoic cavity of 9-day-old SPF chicken embryos cultured with H9N2 virus and then inactivated at 70 °C for 15 min to obtain whole-inactivated H9N2 virus (WIV).

### 4.3. Preparation of Vaccines

The AMP-ZnONPs were prepared via the Borch reaction as previously described [14]. To prepare the H9N2 vaccine formulation, 1 mg/mouse of ZnONPs, 4 mg/mouse of AMP, 4 mg/mouse of AMP-ZnONPs and 5 μg/mouse of CT were mixed separately with 0.2 mL 10^−8.83^ EID_50_/0.1 mL/dose of WIV, and the mixture was incubated for 24 h at 4 °C 10 rpm on a rotary mixer (MX-RD-Pro, DLAB, Beijing, China).

### 4.4. Immunization of Animals

BALB/c mice were orally immunized at 0 days and 14 days. Mice were, respectively, separated into six different vaccine groups (Table 2). Five mice were sacrificed on 21, 28, 35, 42 and 49 days after primary immunization. The schedule of immunization is shown in Figure 1A.

### 4.5. Hemagglutination Inhibition (HI) Assay

Orbital venous blood was collected from mice 21, 28, 35, 42 and 49 days after primary immunization; the serum was isolated by centrifugation (825 g, 10 min) (1–16 K, Sigma, Osterode am Harz, Germany). The antibodies were assessed by HI assay as previously described [24]. For the HI assay, serum samples (25 μL) were diluted in a series of 2-fold dilutions with PBS (25 μL) in 96-well V-bottom plates. Next, 4 haemagglutinating units of the H9N2 virus were added to the diluted serum and incubated at 37 °C for 15 min. Finally, 25 μL of 1% chicken red blood cells were added to the serum–virus mixture and incubated at 37 °C for 15 min.

### 4.6. Determination of Anti-H9N2 Specific Secretory Immunoglobulin A (sIgA), IgG, IgG1 and IgG2a by ELISA Assays

The levels of specific sIgA in intestinal lavage fluid, IgG, IgG1 and IgG2a in serum 21, 28, 35, 42 and 49 days after primary immunization were detected using ELISA (*n* = 5) [24]. In brief, a plate was coated with WIV antigens (100 μL·well^−1^) overnight at 4 °C, which were then blocked for 2 h at 37 °C with 1% BSA. Then, 100 μL of the sample was diluted in 0.1% BSA (sIgA, 1:10; IgG, 1:32; IgG1, 1:10; IgG2a, 1:10) and added to the plate for 2 h at 37 °C. After five washes, 100 μL of HRP-conjugated goat anti-mouse sIgA (1:4000), IgG, IgG1 and IgG2a (1:8000) was incubated on the plate for 1 h. Next, 100 μL of TMB was added for 10 min at 37 °C, and 50 μL of H_2_SO_4_ (2 M) was then added to stop the reaction. The microplate reader (Epoch, BioTek, VT, USA) detected signals at 450 nm.

### 4.7. Activation of the CD4^+^ and CD8^+^ T Cells in Spleen

After the spleen was separated aseptically 28 days after primary immunization, it was treated with RBC lysis buffer and cell density was adjusted to 1 × 10^6^ cells/mL with PBS. T lymphocyte subtypes were detected using fluorescent antibodies including CD3-APC, CD4-FITC and CD8-PE. The cells were incubated with the fluorescent antibodies in the dark at 4 °C for 30 min. After centrifugation, the cells were washed three times with ice-cold PBS and resuspended in 0.5 mL of PBS. The percentages of CD3^+^CD4^+^ T cells and CD3^+^CD8^+^ T cells were analyzed by flow cytometer (CytoFLEX, Beckman, CA, USA).

### 4.8. Immunohistochemical Staining for IgA^+^ Cells

The tissues of the duodenum and jejunum were fixed using formalin. After slicing, the tissue sections are dewaxed to remove the paraffin and treated with 0.6% H_2_O_2_ for 30 min to block any endogenous peroxidase activity. The tissue samples were then incubated with 3% BSA at room temperature for 1 h, and then incubated with the antibody, goat anti-mouse IgA antibody, at a dilution of 1:100, overnight at 4 °C. After washing off any unbound primary antibody, the sections were stained with a peroxidase-containing solution of DAB and H_2_O_2_. This resulted in the development of a brown color where the primary antibody had bound to IgA-positive cells. Finally, the sections were slightly restained with hematoxylin, and a blue-purple stain was used to visualize cell nuclei. Images were obtained with an upright fluorescence microscope (BX53, Olympus, Tokyo, Japan).

### 4.9. Analysis of DCs and B-Cell Subsets by Flow Cytometry

The concentration of mouse mesenteric lymph nodes (MLN) and peyer’s nodule (PP) cells was adjusted to 10^6^/mL. Then, 100 μL cell suspension was prepared. To investigate the effect of vaccine adjuvants on the activation of mouse intestinal DCs, MLN and PP cells were stained with two antibodies: anti-CD11c-eFluor 450 and anti-MHCII-APC. To stain the different subsets of B cells, the antibodies (anti-B220-PerCP-eFluor™ 710 and anti-CD5-FITC) were added to the cell suspension. The cell suspension was stained in the dark at 4 °C for 0.5 h. The suspension was washed with 1 mL PBS and resuspended with 0.5 mL PBS into the flow cytometry tube.

### 4.10. The Expression of mRNA Levels for Cytokines

RNA extraction, cDNA synthesis and qRT-PCR were performed on samples of the jejunum as previously described [14]. Primer sequences for GAPDH, α-chain, J-chain, polymeric immunoglobulin receptor (pIgR), IL-6, TNF-α, B-cell activating factor (BAFF), proliferation-inducing ligand (APRIL), CCR9 and CCL25 are listed in Table 3.

### 4.11. Intestinal Microbiota Community Analysis

The samples were collected immediately after the mice were killed and aseptically dissected, then stored at −80 °C (DW-86L630, Aucma, Qingdao, Shandong Province, China) for DNA extraction and analysis. The V3-V4 region of the 16S rDNA gene was amplified with a common primer pair (F: CCTAYGGGRBGCASCAG; R: GGACTACNNGGGTATCTAAT). High-throughput sequencing analysis of the bacterial rDNA genes was sequenced on the purified and pooled samples using an *Illumina NovaSeq* platform at Suzhou Panomix (Suzhou, Jiangsu Province, China).

### 4.12. Statistical Analysis

All data were analyzed using GraphPad Prism 8.3.0 software (Boston, MA, USA). The results were shown as the mean ± standard deviation (SD), and the statistical differences between groups were evaluated by one-way or two-way ANOVA.

## 5. Conclusions

In summary, our study proposed that AMP-ZnONPs might serve as an innovative adjuvant for orally inactivated H9N2 vaccines by enhancing humoral, cellular and intestinal mucosal immunity. We investigated the action mechanism of AMP-ZnONPs on intestinal immunity by assessing immune cell populations in PP and MLN and changes in gut microbiota. Our results showed that oral WIV + AMP-ZnONPs particularly affected the DCs and B-1 cell population and increased the abundance of beneficial bacteria such as *Lactobacillus* in the gut, providing important insights for the development of advanced adjuvants to enhance the efficacy of next-generation vaccines.

## Figures and Tables

**Figure 1 ijms-25-02132-f001:**
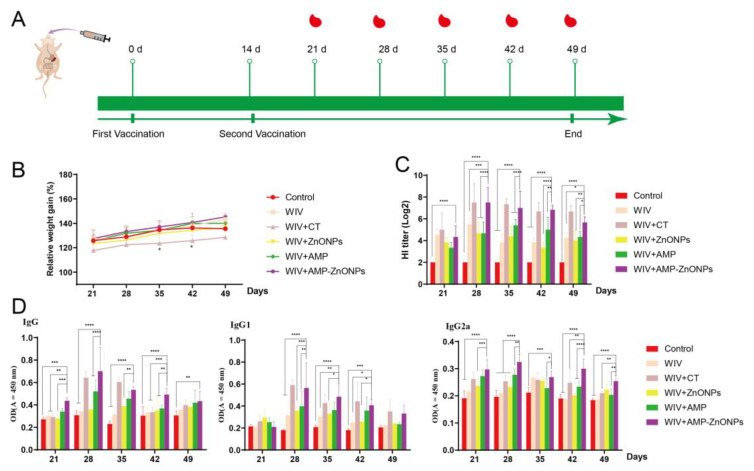
The schedule of immunization strategy (**A**). Changes in body weight of mice over time (**B**). * indicates *p* < 0.05, compared with the control group. HI titers (**C**), IgG, IgG1 and IgG2a (**D**) antibodies in mouse serum on 21, 28, 35, 42, 49 days after primary immunization. *, **, *** and ****, respectively, indicate *p* < 0.05, 0.01, 0.001 and 0.0001, compared with the WIV + AMP-ZnONP group.

**Figure 2 ijms-25-02132-f002:**
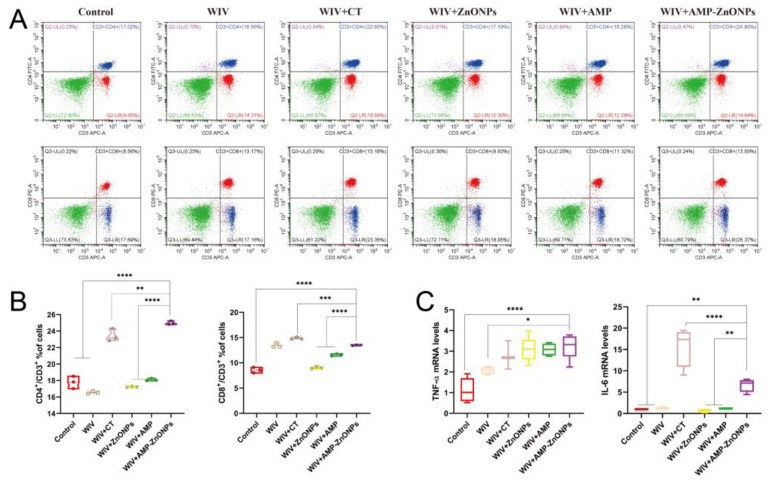
Flow cytometry plots images of CD4^+^/CD3^+^ and CD8^+^/CD3^+^ T cells in various groups (**A**). Statistical analysis of CD4^+^/CD3^+^ and CD8^+^/CD3^+^ ratio (**B**). The levels of cytokines TNF-α and IL-6 in jejunum tissue from immunized mice 28 days after primary immunization were determined by qRT-PCR (**C**). *, **, *** and ****, respectively, indicate *p* < 0.05, 0.01, 0.001 and 0.0001, compared to the WIV + AMP-ZnONP group.

**Figure 3 ijms-25-02132-f003:**
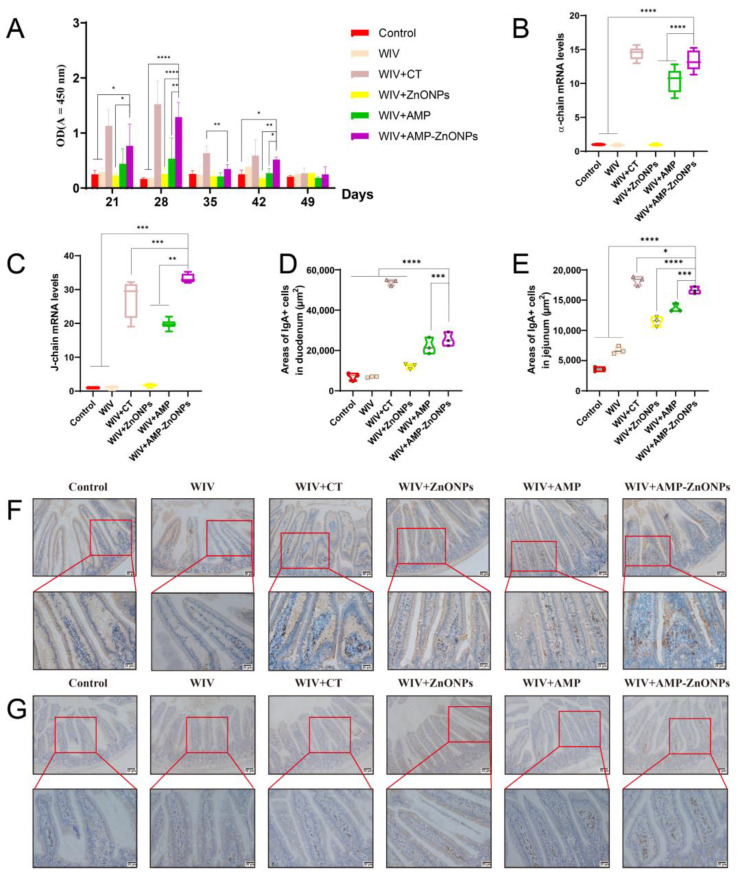
The mucosa IgA titers in the intestinal wash were detected at different time points by ELISA (**A**). The mRNA levels of α-chain (**B**) and J-chain (**C**). Changes in the area of IgA-secreting cells in the duodenum (**D**) and jejunum (**E**). IgA-secreting cells of the duodenum (**F**) and jejunum (**G**) 28 days after primary immunization were observed by IHC. *, **, *** and ****, respectively, indicate *p* < 0.05, 0.01, 0.001 and 0.0001, compared with the WIV + AMP-ZnONP group.

**Figure 4 ijms-25-02132-f004:**
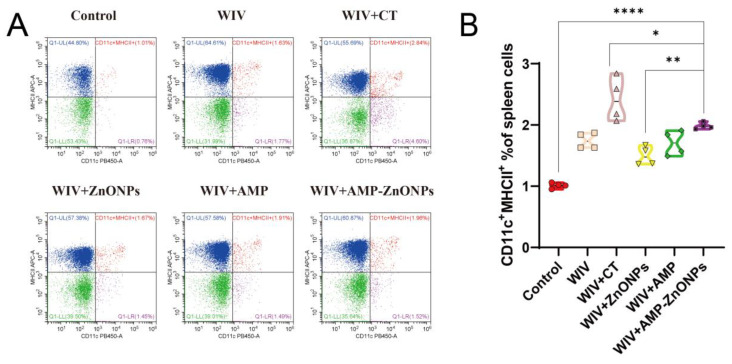
CD11c and MHCII staining were utilized to identify DC subpopulations in the spleen by flow cytometry (**A**). Flow histograms were employed to compare the characteristics of the DC subpopulations (**B**). *, ** and ****, respectively, indicate *p* < 0.05, 0.01 and 0.0001, compared to the WIV + AMP-ZnONP group.

**Figure 5 ijms-25-02132-f005:**
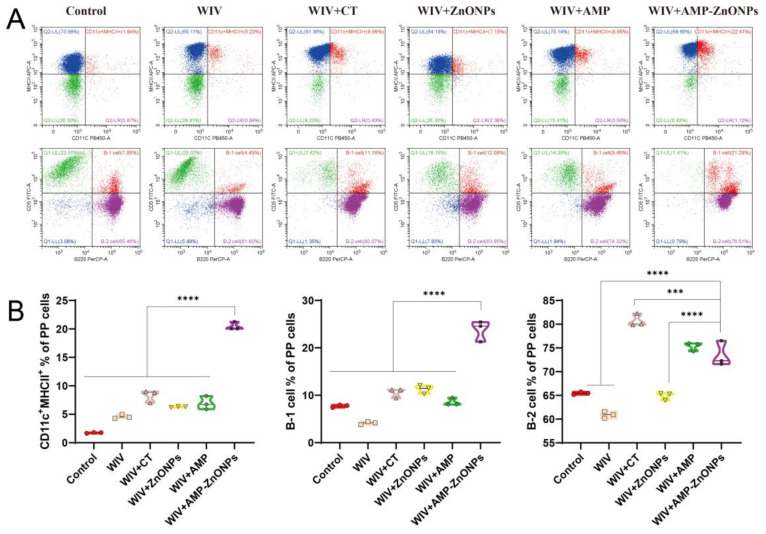
Flow cytometry was performed on single-cell suspensions from PP to identify DC subpopulations using CD11c and MHCII staining, and B-cell subpopulations were identified using CD5 and B220 staining (**A**). Flow histograms were used to compare the DC subpopulations and B-cell subpopulations (**B**). *** and ****, respectively, indicate *p* < 0.001 and 0.0001, compared with the WIV + AMP-ZnONP group.

**Figure 6 ijms-25-02132-f006:**
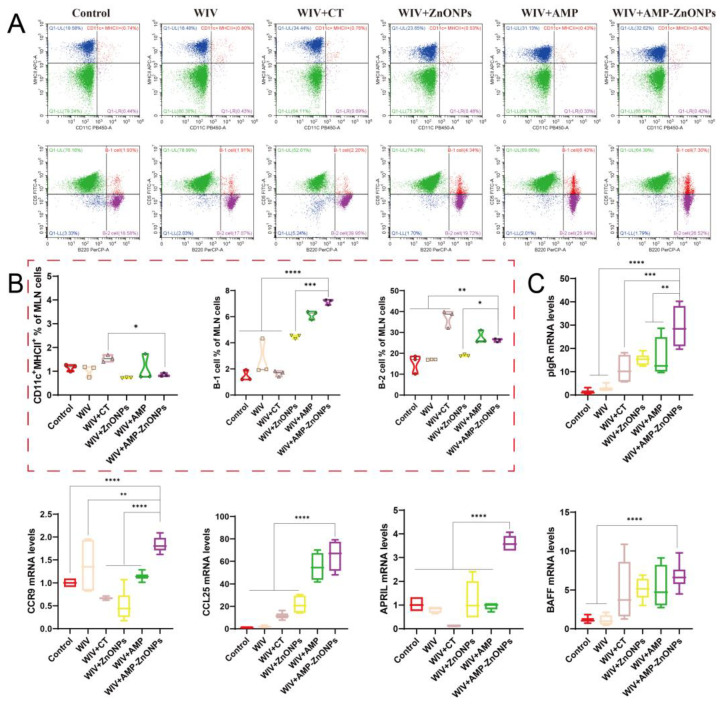
Flow cytometry analysis of DCs and B cells in MLN of immunized mice. The dot plots presented demonstrate the identification of CD11c^+^MHCII^+^ DCs and the numbers of B-1 and B-2 cells (**A**), and they are shown in the corresponding histograms (**B**). The mRNA levels of pIgR, CCR9, CCL25, APRIL and BAFF (**C**). *, **, *** and ****, respectively, indicate *p* < 0.05, 0.01, 0.001 and 0.0001, compared to the WIV + AMP-ZnONP group.

**Figure 7 ijms-25-02132-f007:**
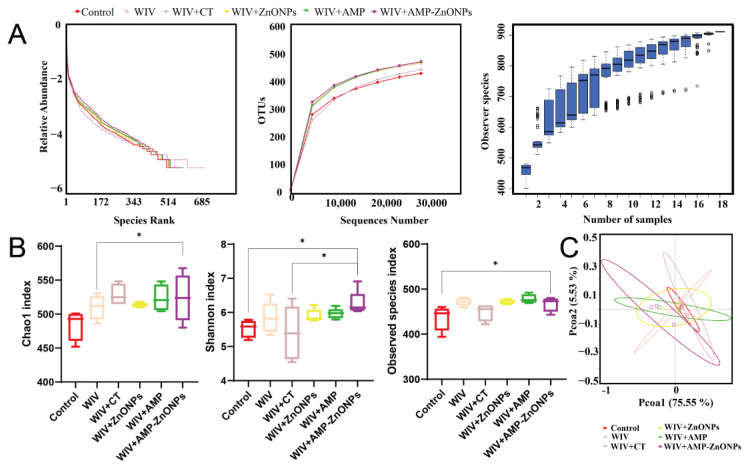
Rank abundance curve, dilution curves and species accumulation boxplot were generated for multiple mouse samples (**A**). The alpha diversity indices including the Chao1 index, Shannon index and observed species index were calculated (**B**). Beta-diversity analyzed by principal coordinates analysis (PCoA) (**C**). * indicates *p* < 0.05, compared with the WIV + AMP-ZnONP group.

**Figure 8 ijms-25-02132-f008:**
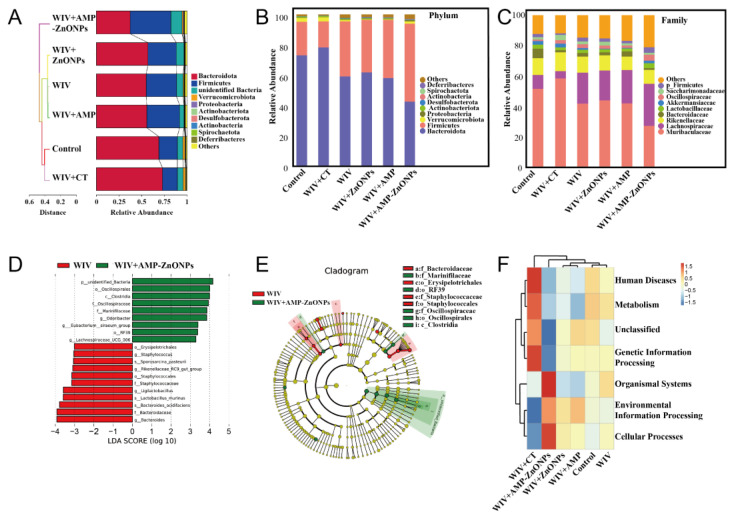
Beta-diversity measured by cluster tree of gut microbiota (**A**). Mean abundances at the phylum (**B**) and family level (**C**) in fecal microbiota of different groups. The LDA scores calculated at the species level between the WIV and WIV + AMP-ZnONP groups (LDA score ≥ 3.0) (**D**). LEfSe evolution branch diagram between WIV and WIV + AMP-ZnONPs (**E**). The cluster heatmap of functional predictive by Tax4Fun (**F**).

**Table 1 ijms-25-02132-t001:** Sequencing data summary for all samples.

Sample Name	Raw PE	Q20	Q30	Effective %
Control 1	93,386	97.67	92.91	80.65
Control 2	82,334	97.77	93.14	75.82
Control 3	95,407	97.64	92.65	74.14
WIV 1	84,102	97.72	92.98	73.55
WIV 2	77,857	97.69	92.93	77.64
WIV 3	54,435	97.98	93.54	76.30
WIV + CT 1	83,070	97.71	92.91	71.07
WIV + CT 2	79,503	97.90	93.51	70.07
WIV + CT 3	85,281	97.97	93.68	77.07
WIV + ZnONPs 1	88,067	97.68	92.95	75.21
WIV + ZnONPs 2	84,474	97.50	92.44	75.76
WIV + ZnONPs 3	84,056	97.62	92.73	77.16
WIV + AMP 1	88,093	97.64	92.80	76.04
WIV + AMP 2	81,627	97.70	92.95	76.70
WIV + AMP 3	86,940	97.74	92.99	71.95
WIV + AMP-ZnONPs 1	87,192	97.76	93.08	83.56
WIV + AMP-ZnONPs 2	86,835	97.83	93.34	86.75
WIV + AMP-ZnONPs 3	87,073	97.73	93.07	75.05

**Table 2 ijms-25-02132-t002:** Grouping of experimental mice and vaccination protocols. (*n* = 25).

Group	Vaccination on d0 and d14	Dose (Per Mouse)
Control	PBS	0.2 mL
WIV + ZnONPs	WIV + ZnONPs	0.2 mL
WIV + AMP	WIV + AMP	0.2 mL
WIV + AMP-ZnONPs	WIV + AMP-ZnONPs	0.2 mL
WIV	WIV	0.2 mL
WIV + CT	WIV + CT	0.2 mL

**Table 3 ijms-25-02132-t003:** Primers of qRT-PCR for the detection of mRNA level.

Gene	Forward Primer	Reverse Primer
GAPDH	ATGGTGAAGGTCGGTGTGAA	CCTTGACTGTGCCGTTGAAT
α-chain	TGAGCGCTGGAACAGTGGCG	TCAGGGCCAGCTCCTCCGAC
J-chain	GGATCCTAATGAGGACATTGTGGAG	CTGGGTGGCAGTAACAACCTGA
pIgR	TCGATGTCAGCCTGGAGGTC	AGGGCATTCAATGGTCACATTTC
IL-6	TTCCATCCAGTTGCCTTCTTG	AATTAAGCCTCCGACTTGTGAA
TNF-α	ATGAGCACAGAAAGCATGATCCGC	AAAGTAGACCTGCCCGGACTC
CCL25	CCGGCATGCTAGGAATTATCA	GGCACTCCTCACGCTTGTACT
CCR9	CTTCAGCTATGACTCCACTGC	CAAGGTGCCCACAATGAACA
APRIL	CTTTCGGTTGCTCTTTGGTTG	CGACAGCACAAGTCACAGC
BAFF	CAGCGACACGCCGACTATAC	CCTCCAAGGCATTTCCTCTTTT

## Data Availability

The data presented in this study are available within the article.

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
