# Peer review of "Polysaccharide from Atractylodes macrocephala Koidz Binding with Zinc Oxide Nanoparticles as a Novel Mucosal Immune Adjuvant for H9N2 Inactivated Vaccine"

_ijms, 2024, doi:10.3390/ijms25042132_

Round 1
Reviewer 1 Report
Comments and Suggestions for Authors
In this manuscript, the authors proposed that AMP-ZnONPs may serve as an innovative adjuvant for orally inactivated H9N2 vaccines by enhancing HI titers, cellular immunity and intestinal mucosal immunity. They investigated the mechanism of action of AMP-ZnONPs on intestinal immunity by assessing immune cell populations in PP and MLN lymph nodes and changes in gut microbiota. Results showed that one oral preparation in particular affected the DCs and B-1 cell population and increased the abundance of beneficial bacteria such as Lactobacillus in the gut. The work provides important insights for the development of advanced adjuvants to enhance the efficacy of next-generation vaccines. I believe this work may be published in IJMS subject to some minor revisions as outlined below:
Line 43-45: this phrase is incoherent
Line 56: exist (plural form of verb needed)
Line 56: ZnOH2+ (the "2+" should be a superscript)
Section 2 should be renamed "Results", as there is a separate Discussion section
Figure 1-7 captions should explain the meaning of the asterisk groupings shown in all these figures.
Table 1: maintain two decimals for all values reported for consistency
Figure 8A-E: for all five panels, figure legends are impossible to read because fonts are way too small
Line 255: "stronger" what? Something is missing. Response probably.
Section 4.1 is incomplete: You should list here all reagent used in all protocols, including and not limited to H2SO4, H2O2, all the reagents needed for the Borch reaction you performed, etc.
Line 370: Indicate g force and manufacturer of centrifuge
Line 375: Indicate manufacturer of incubator
You may want to consider using Figure 9 (which is unusual to have in a Conclusions section) as your graphical abstract instead
References: - add DOIs to all of them
-ensure all journal names are presented in abbreviated format (International Journal of Molecular Sciences is not, for example)
Throughout the manuscript: "et al" - "al" should be followed by a dot like this "al." as it is an abbreviation of a Latin word
Comments on the Quality of English LanguageMinor corrections needed
Author Response
Dear editor and reviewers,
We have read the all comments carefully. Thank the reviewers very much for the constructive evaluations and suggestions concerning our manuscript entitled “Polysaccharide from Atractylodes macrocephala Koidz binding with zinc oxide nanoparticles as a novel mucosal immune adjuvant for H9N2 inactivated vaccine”.
We have checked our manuscript very carefully and tried our best to revise the inaccuracies and mistakes in the whole paper. The responds to reviewer’s comments and suggestions are described in blue below. The corrected sections were marked in red color in the revised manuscript.

Reviewer 2 Report
Comments and Suggestions for Authors
The authors conducted a study on a novel mucosal immune adjuvant for H9N2 inactivated vaccine. The research work is well written and justified through suitable evaluation parameters and references. Though it contains sufficient novelty to be accepted for publication, modifications, and suggestions are recommended to improve the quality of the manuscript. The authors confirmed the efficacy of AMP-ZnONP as an adjuvant of oral H9N2 vaccines. They found that it enhanced not only systemic and mucosal immunity, but also the microflora in the gastrointestinal tract and avoided adverse effects associated with CT. Authors should describe in detail the procedures described in the 'Materials and methods' section. When using an instrument or equipment, indicate the name of its model; manufacturer, city, (state) and country. When using the name of a reagent, indicate the manufacturer, city, state, and country.
The conclusions are consistent. In the discussion section, the authors should explain the strengths and weaknesses of the study.
References are appropriate.
Figure 8 should be clear.
Author Response
Dear reviewer,
We have read all of your comments carefully. Thank you very much for the constructive evaluations and suggestions concerning our manuscript entitled “Polysaccharide from Atractylodes macrocephala Koidz binding with zinc oxide nanoparticles as a novel mucosal immune adjuvant for H9N2 inactivated vaccine”.
We have checked our manuscript very carefully and tried our best to revise the inaccuracies and mistakes in the whole paper. The responds to reviewer’s comments and suggestions are described in blue below. The corrected sections were marked in red color in the revised manuscript.
Detailed responses are attached.

Reviewer 3 Report
Comments and Suggestions for Authors
Dear Authors,
Thank you for an intersting work. After reviewing the manuscript, I have identified specific areas that require corrections and explanations.
1. The manuscript requires grammar and style corrections. I believe that this work would greatly benefit from proofreading by a native speaker.
2. Sentences like “Oral vaccines can reduce adverse stress caused by the injection, such as easy to administer, enhanced mucosal immunity, no need for needles and reduced risk of infection” (lines 43-45) or “Atractylodes macrocephala, an important traditional Chinese medicine, contains polysaccharide from Atractylodes macrocephala Koidz (AMP) as one of its main active components.” (lines 50-52) need to be rewritten. These are only examples of language and style deficiencies. Thus, I strongly suggest proofreading this manuscript.
3. The overall quality of Figures needs to be improved.
4. Line 416 – “qRT-PCR section” – What do you mean by this? If you are referring to previously published work, please correct this. Also, the title “The qRT-PCR for cytokines” is somewhat misleading since you are studying the expression of mRNA levels, not actual cytokines levels.
Comments on the Quality of English LanguageThe text would benefit from some moderate language and style revisions to improve its clarity and coherence.
Author Response
Dear reviewer,
We have read all of your comments carefully. Thank you very much for the constructive evaluations and suggestions concerning our manuscript entitled “Polysaccharide from Atractylodes macrocephala Koidz binding with zinc oxide nanoparticles as a novel mucosal immune adjuvant for H9N2 inactivated vaccine”.
We have checked our manuscript very carefully, and tried our best to revise the inaccuracies and mistakes in the whole paper. The responds to reviewer’s comments and suggestions are described in blue below. The corrected sections were marked in red color in the revised manuscript.
Detailed responses are attached.
